

# The long-term outcomes of deceased-donor liver transplantation for primary biliary cirrhosis: a two-center study in China

Lin Chen[1,*], Xiaodong Shi[2,*], Guoyue Lv[3], Xiaodong Sun[3], Chao Sun[4], Yanjun Cai[1], Junqi Niu[1], Jinglan Jin[1], Ning Liu[5] and Wanyu Li[1]

[1] Department of Hepatology, First Hospital, Jilin University, Changchun, China
[2] Department of Rheumatology, First Hospital, Jilin University, Changchun, China
[3] Department of Hepatobiliary Pancreatic Surgery, First Hospital, Jilin University, Changchun, China
[4] Department of Transplant Center, First Central Hospital, Tianjin, Tianjin, China
[5] Department of Cardiology, First Hospital, Jilin University, Changchun, China
[*] These authors contributed equally to this work.

## ABSTRACT

**Background & Aims**. Factors that influence the outcomes after deceased-donor liver transplantation (DDLT) for primary biliary cirrhosis (PBC) are not well known. We aimed to clarify these effects on the outcomes after DDLT.

**Methods**. We retrospectively analyzed patients with PBC who underwent DDLT from March 2006 to July 2018 at the organ transplantation center of the First Hospital of Jilin University and the First Central Hospital of Tianjin. Changes in liver function were assessed posttransplantation. Recurrence, survival rate, and complications were recorded at follow-up. The effect of liver transplantation on survival and recurrence was evaluated using univariate and/or multivariate Cox regression analyses.

**Results**. In total, 69 patients with PBC undergoing DDLT were included in this study. At 4 weeks posttransplant, all liver function tests were normal. During a median follow-up time of 32 months, 5-year overall survival and recurrence rates were estimated as 95.1% and 21.8%, respectively. A recipient aspartate aminotransferase-to-platelet ratio index (APRI) greater than 2 was negatively associated with survival ($P = 0.0018$). Multivariate regression analysis demonstrated that age younger than 48 years was an independent risk factor for recurrent PBC in recipients undergoing liver transplantation (hazard ratio 0.028, 95% confidence interval 0.01–0.71, $P = 0.03$). Posttransplant infections (62%) and biliary tract complications (26%) were the most common complications.

**Conclusion**. Liver transplantation is an effective treatment for patients with PBC. Liver function normalizes by 4 weeks posttransplant. Although posttransplant survival rate is high, recurrence is possible. To some extent, survival rate and recurrence rate can be predicted by APRI and age, respectively.

Corresponding author
Wanyu Li, liwanyu2006@163.com

## INTRODUCTION

Primary biliary cirrhosis (PBC) is a chronic cholestatic liver disease characterized by the immune-mediated destruction of intrahepatic bile ducts, resulting in fibrosis, cirrhosis, and ultimately, liver failure (*Lindor, 2007*). PBC primarily affects middle-aged women and is diagnosed by the presence of biochemical markers for cholestasis and highly specific serological antimitochondrial antibodies (AMAs) as well as histologically progressive, nonsuppurative, destructive cholangitis affecting small and medium bile ducts (*Selmi et al., 2011*). Without therapy, the median survival of patients with PBC is reported as 7.5 years in symptomatic patients and 16 years in asymptomatic patients (*Lee & Kaplan, 2005*). The use of ursodeoxycholic acid (UDCA) is effective in improving liver function, slows the progression of liver fibrosis, and significantly reduces the need for liver transplantation. Up to 40% of patients with PBC do not adequately respond to UDCA monotherapy (*Akamatsu & Sugawara, 2012*). Obeticholic acid (OCA) as a second-line treatment for PBC patients with incomplete ursodeoxycholic acid response; however, approximately 50% of patients might need other treatments to achieve treatment goals (*Shah & Kowdley, 2019*). Therefore, liver transplantation is the only effective treatment for patients with advanced PBC.

When compared with large European series, where PBC constituted about 9% of all liver transplantation (*Schoning et al., 2015*), PBC accounts for a relatively low proportion (3.5%) in this Asian series (*Sun et al., 2011*). The prognosis of liver transplantation for patients with PBC is favorable compared with other indications. In Western countries, overall patient survival from liver transplantation for PBC was 93–94% at 1 year, 90–91% at 3 years, and 82–86% at 5 years (*Liberal et al., 2013*). However, posttransplant survival rates for alcohol-associated liver cirrhosis and hepatic failure are lower than those for PBC (86%, 73%, and 59% for alcohol-associated liver cirrhosis and 70%, 64%, and 58% for hepatic failure at 1, 5, and 10 years, respectively) (*Adam et al., 2012*).

Recurrent PBC (rPBC) after liver transplantation was first documented in 1982 (*Neuberger et al., 1982*). With the development of surgical techniques, transplant organ storage, and immunosuppressants, the posttransplant survival rate for PBC has improved. However, with extensions in survival, rPBC is not uncommon. In a Japanese multicenter study with 444 patients with PBC, relapse rates of PBC posttransplant were 9.6%, 20.6%, and 40.4% at 5, 10, and 15 years, respectively (*Egawa et al., 2016*). Recurrent disease itself rarely affects patient and graft survival in short- and medium-term follow-up. However, in the long term, rPBC may have a negative impact on patient and graft survival. One study reports that recurrent PBC results in graft loss at a rate of 1.0–5.4% (*Carrion & Bhamidimarri, 2013*). Despite improved survival rates, certain complications can still affect posttransplant prognosis. Our study aimed to report the long-term outcomes of deceased-donor liver transplantation (DDLT) in patients with PBC in northern China.

## PATIENTS AND METHODS

### Patients

We retrospectively studied data from all 69 patients who underwent DDLT for PBC from March 2006 to July 2018 at the organ transplantation center of the First Hospital
of Jilin University and the First Central Hospital of Tianjin. The diagnosis of PBC is based on established diagnostic criteria of *EASL (2017)*. The 69 patients received UDCA therapy (13–15 mg/kg/day) pre- and post-transplant, and had no OCA and immunosuppressisve treatment preoperatively. The AMAs of 69 patients were all postive before liver transplantation. We declare that this study has been approved by the Ethics Committee of the First Hospital of Jilin University and the First Central Hospital of Tianjin, written informed consent was obtained from all participants.

## Calculation formulas

The model for end-stage liver disease (MELD) score was calculated using the following formula: $R = 3.8*\ln$ (bilirubin in mg/dl)$+11.2*\ln$ (international normalized ratio)$+9.6$ $\ln$ (creatinine in mg/dl)$+6.4*$ etiology (etiology:0 for cholestatic or alcohol-related liver disease; 1 for others). And, the formula for aspartate aminotransferase-to-platelet ratio index (APRI): R= (aspartate aminotransferase/ the upper limits of normal)*100/(platelet count in 10 ∧9/L). The updated Mayo risk score (*Murtaugh et al., 1994*) was calculated by the formula: $R = 6.843+0.051*[age] +1.029*\ln$(bilirubin in mg/dl)$+1.020**\ln$ (international normalized ratio)$-3.304*\ln$(albumin in g/dl)$+0.675*$ (edema score)(edema score:0 for no edema and diuretic therapy, 0.5 for edema present without diuretic or edema eliminated by diuretic therapy, 1.0 for edema despite diuretic therapy).

## Histologic examination

Liver biopsy was performed when patients had unexplainable abnormal liver function post-transplant. The diagnosis of acute transplant rejection and chronic transplant rejection were based on the international Banff schema (*Demetris et al., 2000*). Recurrent PBC was diagnosed when either of the following findings were observed: (i) granulomatous cholangitis or epithelioid granulomas without any sign of infection or (ii) dense lymphoid aggregates in the portal tracts not associated with perivenular inflammation or endotheliitis (*Haga et al., 2007*).

## Serologic liver function

For each patient, liver function tests [aspartate aminotransferase (AST), alanine aminotransferase (ALT), alkaline phosphatase (ALP), gamma-glutamyl transpeptidase (GGT), total bilirubin (TBIL), direct bilirubin (DBIL), albumin (ALB)] were performed immediately before the transplant, daily for 4 days after the transplant, and weekly for 4 weeks posttransplant. If AST or/and ALT was greater than 1500 U/L at 1 week after transplant, it was classified as initial poor graft function (IPGF) (*Kocbiyik et al., 2009*). Of the initial 69 patients, 5 patients were excluded from the analysis of liver function results (5 due to uncompleted data). In total, the liver function results of 64 patients were analyzed.

## Diagnosis of complications

Complications were diagnosed based on symptoms, laboratory results, and radiological findings. Infection-related complications were diagnosed based on established diagnostic criteria (*Horan, Andrus & Dudeck, 2008*). Infections can be classified by anatomy (such as respiratory system, renal system, biliary system) or by microbiology (such as bacteria,
fungus, virus). In this study, most infection complications were based on diagnostic microbiology; some infection complications with negative diagnostic microbiology were diagnosed based on clinical presentation and radiology.

Biliary complications included strictures (anastomotic and nonanastomotic), leakage, and bile tumors. Anastomotic stricture was diagnosed as a segmental stricture located within one cm of the biliary anastomosis as shown by magnetic resonance cholangiopancreatography (MRCP), endoscopic retrograde cholangiopancreatography (ERCP), or percutaneous cholangiography. Ischemic-type biliary lesions were diagnosed when either MRCP or ERCP showed typical signs of segmental or diffuse intrahepatic or extrahepatic strictures or biliary tree destruction. Biliary leakage was determined by abdominal fluid drainage tube or bile drainage under ultrasound-guided puncture. Color Doppler ultrasound was performed the second day after liver transplant or with abnormal liver function. Abdominal computed tomography with contrast or angiography was performed when complications of the portal vein or hepatic artery were detected by color Doppler ultrasound.

Diabetes mellitus was defined as a fasting glucose higher than seven mmol/L or a random glucose higher than 11.1 mmol/L and a hemoglobin A1C higher than 6.5%.

### Statistical analysis

Descriptive statistics are presented as median and range for continuous variables and counts and percentages for categorical variables. Overall survival and PBC recurrence-free curves were estimated according to the Kaplan–Meier method. The log-rank test was used to evaluate whether each variable significantly affected postoperative patient survival and PBC recurrence. The hazard ratio and 95% confidence interval were calculated by multivariate Cox regression analysis. A $P$-value of 0.05 was considered to be statistically significant.

## RESULTS

### Patient characteristic

In our study, the follow-up period ranged from 0.5 to 107 (median, 32) months. The baseline characteristics of the 69 patients are summarized in Table 1. The median age was 55 years (range, 39–67), and 60 patients (87%) were female. The MELD and APRI were 19 (range, 6–31) and 2.26 (range, 0.11–42.22), respectively. The median updated Mayo risk score was 9.53 (range, 4.34–12.44). The median levels of AST and ALT were 75 (range, 15–513) and 45 (range, 6-229), respectively. The median ALP and GGT levels were 186.5 (range, 54–732)and 93.5 IU/L (range, 14–779), respectively. The median value of TBIL was 9.83 (range, 0.41–41.6 mg/dl). The median value of ALB was 3 g/dl (range, 2.3–4).

Of these cases, 26 patients were classified as Child-Pugh grade B; 43 patients, grade C. One patient had a second liver transplant due to thrombosis of the hepatic artery 2 weeks after the first liver transplant.

**Table 1** Baseline characteristics.

|  | No.(percent) or median(range) |
| --- | --- |
| Age(Y) | 55(39-67) |
| Gender | |
|     Female | 60(87.0%) |
|     Male | 9(13.0%) |
| Follow-up period(months) | 32.0(0.5–107) |
| APRI | 2.26(0.11-42.22) |
| MELD | 19(6-31) |
| Updated Mayo risk score | 9.53(4.34–12.44) |
| Child-pugh grade | |
|     B | 26(37.7%) |
|     C | 43(62.3%) |
| Preoperative liver function | |
|     AST (IU/L) | 75(15–513) |
|     ALT (IU/L) | 45(6–229) |
|     ALP (IU/L) | 186.5(54–732) |
|     GGT (IU/L) | 93.5(14–779) |
|     TBIL (mg/dl) | 9.83(0.41–41.6) |
|     ALB (g/dl) | 3.0(2.3–4.4) |

**Notes.**

APRI, aspartate aminotransferase-to-platelet ratio index; MELD, model for end-stage liver disease score; AST, aspartate aminotransferase; ALT, alanine aminotransferase; ALP, alkaline phosphatase; GGT, gamma- glutamyl transpeptidase; TBIL, total bilirubin; ALB, Albumin.

Normal range: AST 13-35, ALT 7-40, ALP 35-125, GGT 7-45, TBIL 0.40-1.75, ALB4-5.5.

### APRI greater than 2 associated with poor survival of PBC patient survival after liver transplantation

During the follow-up, three patients died. One died from abdominal hemorrhage after transplant, and the other two died from infection and liver failure at 14 months posttransplantation. Overall patient survival in the study population was 98.6% at 1 year, 95.1% at 3 years, and 95.1% at 5 years after transplantation (Fig. 1). Multivariate Cox regression analysis were failed to perform due to the small samples of death.A log-rank test of the Kaplan–Meier life table analysis showed that a pre-transplant APRI greater than 2 was related to poor survival ($P = 0.0018$) (Fig. 2). The Kaplan–Meier life table analysis with a log-rank test failed to find that age, gender, the MELD score, Child-Pugh score, Mayo risk score , pre- and post- operative liver function were statistically significant in predicting the survival of liver transplantation.

### Younger age at liver transplantation associated with a greater risk for recurrent PBC

Of 69 patients, 7 patients who had unexplained abnormal liver function underwent percutaneous liver biopsy postoperatively. Only 1 patient was diagnosed rejection reaction, and another 6 patients were diagnosed recurrent PBC. The incidence of recurrence increased with time after transplantation (Fig. 3), leading to relapse rates of 3.5% at 1 year,

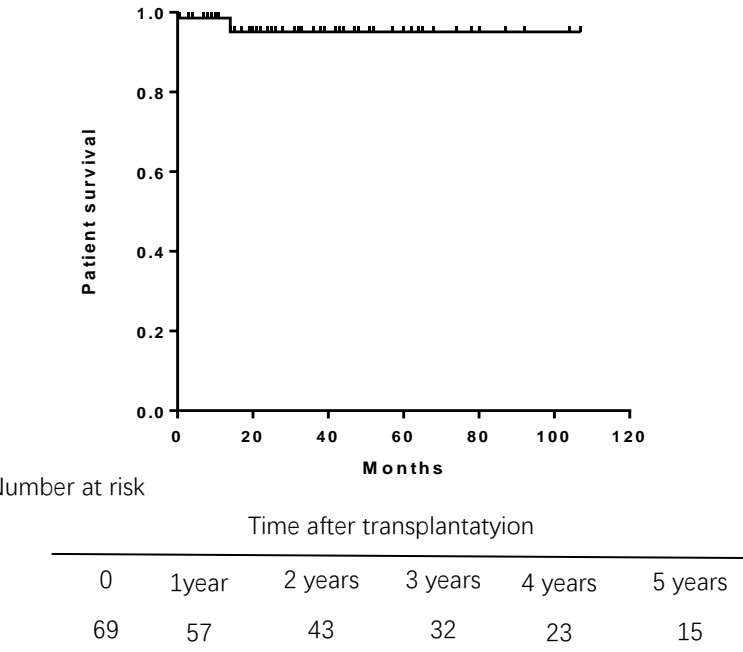

**Figure 1 Overall survival rate of 69 patients of PBC after liver transplantation.** Overall survival rate was 98.6% at 1 year, 95.1% at 3 years. PBC, primary biliary cirrhosis.

8.1% at 3 years, and 21.8% at 5 years after transplantation. The median duration from liver transplantation to the diagnosis of recurrent was 21.5 months. The Kaplan–Meier life table analysis with a log-rank test revealed a preoperative factor of recipients aged 48 years or younger as the only significant risk factor of disease recurrence (Fig. 4). In a multivariate Cox regression analyse model adjusted for gender, pre-operative Mayo risk score, APRI score, MELD score, a preoperative age of 48 years or younger remained the only independent risk factor of patient recurrence (hazard ratio 0.028, 95% confidence interval 0.01–0.71, $P = 0.03$)

## Normal liver function at 4 weeks posttransplantation

The curves of AST and ALT changes posttransplantion were similar—reaching a peak on the first day, dramatically decreasing on the second day (AST declined more than ALT), returning to normal by 1 week, and remaining stable with normal levels at 4 weeks (Fig. 5A). The curves of ALP and GGT changes were also somewhat similar after transplant (Fig. 5B)—both returned to normal on the first day but then slowly increased. However, ALP peaked at the seventh day and then returned to the normal range; GGT peaked at 2 weeks (2 times higher than the upper cutoff) and then slowly decreased to normal by 4 weeks.

The levels of TBIL and direct bilirubin (DBIL)decreased after transplantation to normal by 4 weeks (Fig. 5C). The level of ALB eventually increased to normal by 4 weeks (Fig. 5D). The peak levels of ALT and/or AST were greater than 1500 U/L in 15 patients. The frequency of IPGF was 21.7%.

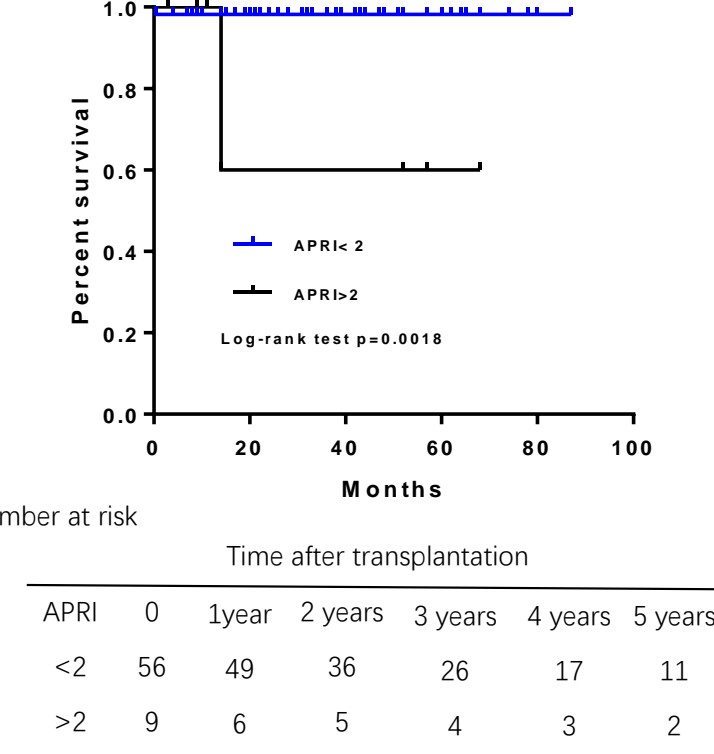

Number at risk

Time after transplantation

| APRI | 0 | 1year | 2 years | 3 years | 4 years | 5 years |
|------|------|-------|---------|---------|---------|---------|
| <2 | 56 | 49 | 36 | 26 | 17 | 11 |
| >2 | 9 | 6 | 5 | 4 | 3 | 2 |

**Figure 2  Patient survival rate according to the APRI before liver transplantation.** APRI greater than 2 had a shorter overall survival.

## Infection and biliary complications

Postoperative complications of the 69 patients are summarized in Table 2. A total of 43 patients (62%) developed 63 episodes of infections in our follow-up period. Of these patients, 16 developed multiple infectious complications. The vast majority of infections were microbiological infection. Among the cases of infection without evidence of microbiology, there were four cases of pneumonia, one upper respiratory infection, one bile duct infection, and one urinary infection. The 56 microbiologically defined episodes were categorized by the following causes: 42 (75%), bacteria; 6 (11%), fungus; 6 (11%), mixture of bacteria and fungus; and 2 (4%), virus.

During the follow-up, 18 (26%) patients had biliary complications, including 7 anastomotic strictures (10%), 10 nonanastomotic strictures (14%), 3 leakage (4.3%), and 1 biloma (1.5%). One patient had biliary leakage and nonanastomotic strictures; two patients had anastomotic strictures secondary to leakage. Among the five patients with vascular complications, two(2.9%) had portal vein thrombosis, and three(4.3%) had hepatic artery thrombosis. One patient had a second liver transplant due to liver failure caused by nonanastomotic strictures secondary to hepatic artery thrombosis. Only one (69) patient had a rejection reaction, which occurred at 33 months. Other complications included osteoporosis (5), hepatitis B infection (3), and diabetes (2).

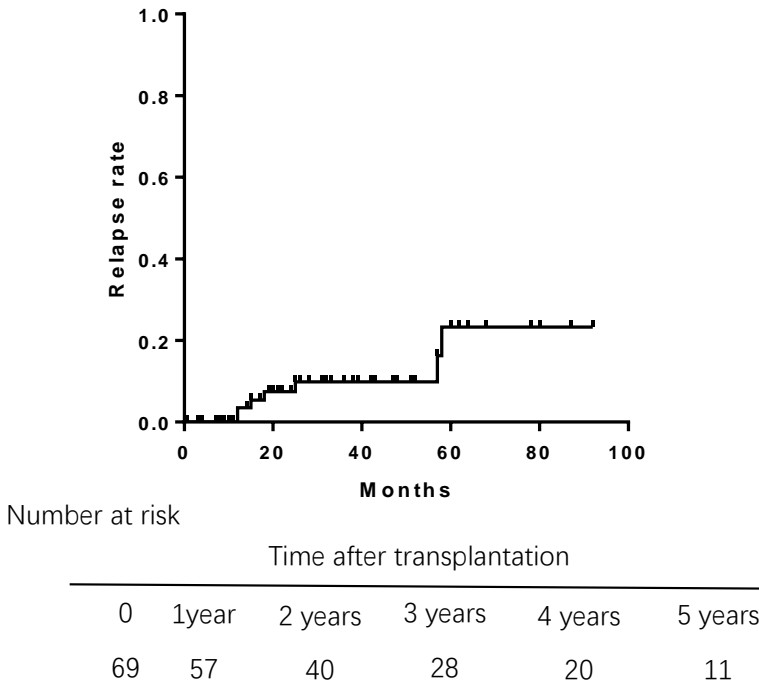

Number at risk

Time after transplantation

| 0 | 1year | 2 years | 3 years | 4 years | 5 years |
|---|---|---|---|---|---|
| 69 | 57 | 40 | 28 | 20 | 11 |

**Figure 3** **Rate of relapse of PBC aftre liver transplantation.** The relapse rates of PBC was 3.5% at 1 year, 8.1% at 3 years, and 21.8% at 5 years after transplantation.

## DISCUSSION

Prognosis is favorable for patients with PBC after liver transplantation. Among our patients who underwent DDLT for PBC, the causes of mortality were abdominal hemorrhage, infection and liver failure. The 1- and 3-year survival rates were 98.6% and 95.1%, respectively, after liver transplantation. In a retrospective study of the United Network for Organ Sharing database, patient survival at 1, 3, and 5 years for living donor liver transplantaion was 95.5%, 93.6%, and 92.5% and for dead donor liver transplantation was 90.9%, 86.5%, and 84.9%, respectively (*Kashyap et al., 2010*). Compared with previously reported outcomes, overall survival was higher in our series. Several inconsistencies remain regarding factors prognostic of patient survival after liver transplantation among various cohorts. Similar to Bhat's research, we found that an APRI greater than 2 is a negative predictor of patient survival after liver transplantation for PBC (*Bhat et al., 2015*). Therefore, APRI could be of clinical use to predict survival rate.

AMA is not a good marker for rPBC because it frequently remains positive after transplantation (*Dubel et al., 1995*). Thus, liver puncture biopsy is the gold diagnostic standard for rPBC. Due to different diagnostic criteria and the use of protocol biopsies by transplant programs, the reported relapse rates of PBC range from 1% to 53% (*Akamatsu & Sugawara, 2012*; *Bosch et al., 2015*). In our study, recurrence rate increased after transplant as recurrence risks grew: 3.5% at 1 year, 8.1% at 3 years, and 21.8% at 5 years. These findings are consistent with those of previous studies (*Charatcharoenwitthaya et al., 2007*).

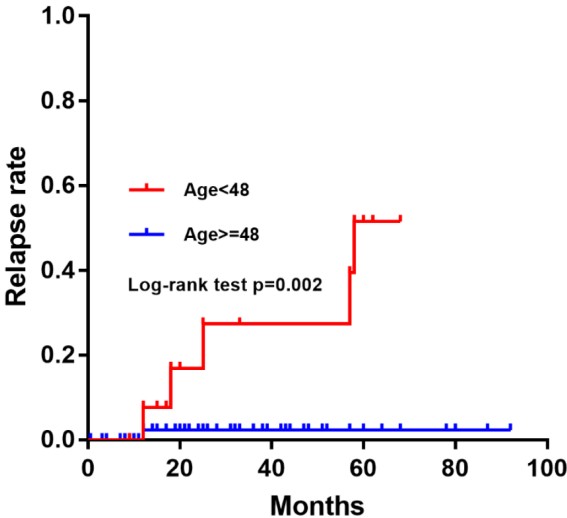

**Number at risk**

| | | Time after transplantation | | | | |
|---|---|---|---|---|---|---|
| Age | 0 | 1year | 2 years | 3 years | 4 years | 5 years |
| <48 | 14 | 12 | 8 | 6 | 6 | 3 |
| ⩾48 | 55 | 43 | 31 | 22 | 12 | 6 |

**Figure 4** **Comparision of the rate of recurrent PBC among different age groups at liver transplantation.** Those aged 48 years or younger had high relapse rates of PBC.

However, other studies found a higher recurrence risk when asymptomatic patients had regular protocol biopsies. Unlike biopsies performed only after identifying liver dysfunction, regular protocol biopsies can detect potential recurrence before dysfunction occurs (*Charatcharoenwitthaya et al., 2007*). Therefore, more rPBCs can be detected by protocal biopsies. However, the effect of recurrent disease on the outcome of patients who performed liver transplantaion for PBC is uncertain (*Akamatsu & Sugawara, 2012*), whether patients can benefit from protocal biopsies needs further study. Previous studies have identified several factors associated with rPBC, including age, number of HLA mismatches, and the use of immunosuppressants. However, the association between age and recurrence is still being debated (*Charatcharoenwitthaya et al., 2007*). Consistent with our study's findings, a Japanese study indicated that recipients younger than 48 years are at risk for recurrence (*Egawa et al., 2016*). However, the mechanism involved remains unknown.

After liver transplantation, liver function recovered by 4 weeks; ALT and AST activities each peaked once before this recovery. In a retrospective study of 75 liver transplant patients, transaminase activities reached 10–12 times above the upper limits of normal on the first day after surgery and reached a normal range by the ninth day, similar to our results. These findings suggest that the early peaks of transaminase activity represent preservative damage during liver removal from the donor and subsequent ischemia-reperfusion injury during

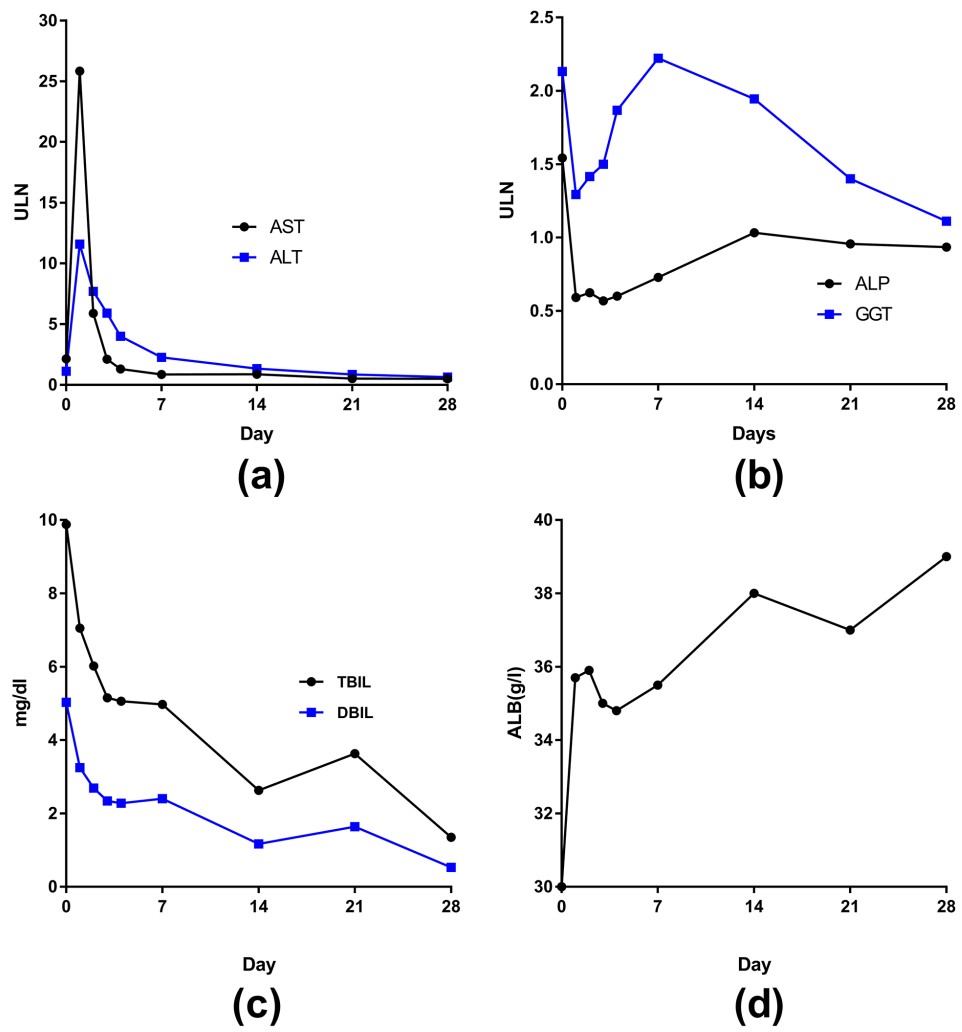

**Figure 5  The curves of median liver function in 62 liver transplant patients changed posttransplantion.** (A) AST and ALT; (B) ALP and GGT; (C) TBIL and DBIL; (D) ALB.

transplantation (*Naik et al., 2013*). In our study, the frequency of IGPF was 21.7%, which was lower than that in other studies (29.5%) (*Chui et al., 2000*). Different from the ALP curve, GGT levels increased until peaking at the seventh day, whereas ALP levels increased until peaking at the second week, and in both cases the levels then decreased until reaching the normal range by the first month. A previous study reports that GGT levels increase to a maximum at day 9 after surgery and then decrease. This previous study showed that the post-operative GGT level was significantly associated with the survival of liver transplantation. Our study failed to find that pre/post-operative GGT level was statistically associated with the outcome of liver transplantation. The early elevation in GGT after liver transplantation may produced by ischemia-reperfusion and surgical stress in the recipient (*Zhang et al., 2007*). The levels of TBIL and DBIL decreased after transplantation to normal by 4 weeks. Bilirubin is a distinct predictive biomarker for liver function recovery. Dynamic monitoring of bilirubin is helpful for early detection of biliary complications.

**Table 2  Postoperative complications.**

| | |
|---|---|
| Infection complications | 63(63%) |
| Respiratory infection | 30 |
| Biliary infection | 13 |
| Bacteremia | 6 |
| Urinary tract infection | 5 |
| Abdominal infection | 4 |
| Skin or tissue infection | 1 |
| Gastrointestinal infection | 1 |
| Thoracic infection | 1 |
| Cytomegalovirus infection | 1 |
| Biliary complications | 21(21%) |
| Ischemic-type biliary lesions | 10 |
| Anastomotic stricture | 7 |
| Bile leakage | 3 |
| Biloma | 1 |
| Vascular complications | 5(5%) |
| Hepatic artery thrombosis | 3 |
| Portal vein thrombosis | 2 |
| Rejection | 1(1%) |
| Chronic rejection | 1 |
| Other complications | 10(10%) |
| Osteoporosis | 5 |
| De novo hepatitis B infection | 3 |
| Diabetes mellitus | 2 |

Despite advances in surgical techniques, organ preservation, immunosuppression, and infection prevention, various complications still affect a patient's prognosis, including infections, biliary complications, vascular complications, and rejections. In our study, the frequencies of infection and biliary complications were 62% and 21%, respectively. Bacterial infection was the most common complication, consistent with other studies (*Li et al., 2012*; *Vera, Contreras & Guevara, 2011*). Thus, the biggest problem is controlling infection to decrease posttransplant complications. Biliary complications are the primary morbidity in liver transplant patients, have an incidence of 10–30%, and even cause mortality (*Mejia et al., 2016*; *Wojcicki, Milkiewicz & Silva, 2008*), consistent with the 26% incidence of biliary tract in our study. The similarity of these results suggests that even though PBC affects small bile ducts, biliary complications have not dramatically increased. Timely diagnosis and aggressive treatment are vital for the survival of patients and grafts (*Khalaf, 2010*). Thrombosis of the hepatic artery is a fatal vascular complication with a rate of 3–5% and usually requires re-transplantation, with a high mortality incidence of 20–60% (*Khalaf, 2010*; *Ma, Lu & Luo, 2016*). In our study, 3 (4.3%) patients had thromboses of the hepatic artery, and 2 (2.9%) had portal vein thrombosis, consistent with previous studies (*Khalaf, 2010*).

One limitation in this study is the review of liver pathology. Liver biopsy was performed when patients had unexplainable abnormal liver function rather than protocol biopsies in this study, so early recurrence of PBC in our patients may have been missed. In future researches, protocol criteria may detect early recurrence timely. Another limitation is that the small samples of patients does not allow multivariate analysis of survival. In the future, when the the number of sample increases, we will further analyze the risk factors of survival.

## CONCLUSIONS

Liver transplantation is an effective treatment for patients with PBC due to its excellent postoperative survival rate. After transplantation, liver function recovers well with decreased infection and biliary complications. With advances in the technique, the survival rate after transplantation has become very high, but rPBC is not rare and is related to patient age. To some extent, APRI can be used to predict a patient's survival.

### Funding
This study was supported by grants from the First Hospital of Jilin University (NO. JDYYJC006), the Tianqing Liver Disease Research Fund (NO. TQGB20180160), and the Department of Science and Technology of Jilin Province (NO.20180520116JH). The funders had no role in study design, data collection and analysis, decision to publish, or preparation of the manuscript.

### Grant Disclosures
The following grant information was disclosed by the authors:
First Hospital of Jilin University: JDYYJC006.
Tianqing Liver Disease Research Fund: TQGB20180160.
Department of Science and Technology of Jilin Province: 20180520116JH.

### Competing Interests
The authors declare there are no competing interests.

### Author Contributions
- Lin Chen and Xiaodong Shi conceived and designed the experiments, performed the experiments, analyzed the data, prepared figures and/or tables, authored or reviewed drafts of the paper, and approved the final draft.
- Guoyue Lv and Xiaodong Sun performed the experiments, analyzed the data, prepared figures and/or tables, and approved the final draft.
- Chao Sun and Yanjun Cai analyzed the data, prepared figures and/or tables, authored or reviewed drafts of the paper, and approved the final draft.
- Junqi Niu and Jinglan Jin conceived and designed the experiments, authored or reviewed drafts of the paper, and approved the final draft.

- Ning Liu analyzed the data, prepared figures and/or tables, and approved the final draft.
- Wanyu Li conceived and designed the experiments, performed the experiments, authored or reviewed drafts of the paper, and approved the final draft.

## Human Ethics

The following information was supplied relating to ethical approvals (i.e., approving body and any reference numbers):

We declare that this study has been approved by the Ethics Committee of the First Hospital of Jilin University and the First Central Hospital of Tianjin.

## Data Availability

Raw data is available in the Supplementary Files.

## Supplemental Information

Supplemental information for this article can be found online at http://dx.doi.org/10.7717/peerj.9563#supplemental-information.

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
