# Peer review of "The long-term outcomes of deceased-donor liver transplantation for primary biliary cirrhosis: a two-center study in China"

_PeerJ, doi:10.7717/peerj.9563_

## Round 0.1 · original submission · Major Revisions

I am sorry for the long delay in getting your manuscript reviewed but it proved difficult to find appropriate reviewers. Given the current situation, do not consider the 55-day resubmission deadline as mandatory.

Reviewer 1 ·

Basic reporting

Chen et al. analyzed the outcome for primary biliary cirrhosis (PBC) up to 5 years after liver transplantation. The authors’ main findings are that a recipient aminotransferase-to-platelet ratio index (APRI) under 2 showed significantly better overall survival and that an age under 50 showed significantly higher risk of PBC recurrence.

I. Introduction
This chapter is well-written, and the references are all pertinent. However, in line 89, “alcoholic cirrhosis” should be changed to alcohol-related or -associated liver cirrhosis. The American Association for the Study of Liver Disease (AASLD) and the European Association for the Study of the Liver (EASL) have updated their guidelines on alcohol-related liver disease (reference, Crabb DW et al. Hepatology. 2020. EASL. J Hepatol. 2018). One of the most significant updates is a change from the term “alcoholic” to “alcohol-related (or-associated).” Even though the name of a disease is a relatively superficial concern, in this case, the word “alcoholic” is very stigmatizing. Therefore, the phrase “alcohol-related” is suggested instead. In addition, the phrase “liver failure” is vague. Please employ a more specific term.

II. Table 1
Information about the full terms abbreviated as AST, ALT, ALP, GGT, T-Bil, and ALB would be helpful.

Experimental design

I. Patients and methods
• Line 106: Provide details about the type of liver transplantation (living-donor liver transplantation vs. deceased-donor liver transplantation).
• Patients and Methods: Indicate the formulas for APRI and MELD.

II. Results
• The authors focused on APRI as a prognostic tool for post-transplantation outcomes. Please clarify the reason why the authors focused only on APRI and not other fibrosis markers.
• The authors classified patients by APRI and age. However, it is not clear why the cutoff values for these were set to 2 and 50, respectively. Additionally, information about both hazard ratio and the 95% confidence interval in the log-rank tests is missing. Univariate analysis of overall survival for the 62 patients, including gender and incorporating pre-transplant age, Child-Pugh score, MELD score, aspartate aminotransferase, alanine aminotransferase, and platelet count, should be performed.

Validity of the findings

no comment.

Reviewer 2 ·

Basic reporting

English writing at a high standard and reasonable literature references etc. However major issues include;

1) An abstract which is unstructured.

2) Poor structure: methods section and results section seem to be blended, rather than separated (line 105 to 127 in mostly made up of results, rather than methods).

3) the term "median end-stage liver disease score" is incorrect, it should be "model for End-stage Liver Disease score" and abbreviated as MELD.

4) In Figure 4, it should be clarified if LFTs are mean or median.

5) in line 178, it should be clarified if APRI score is pre-transplant or post-transplant.

6) in line 187, the term "negative indicator" is not a scientific conventional term. Please rephrase.

7) line 203, "The vast majority of infections were microbiologically y bile"- what does "microbiologically y" mean? It should be deleted.

8) line 211-212 should have percentages after "two" and "three"

Experimental design

Major issues include:

1) It is unclear when the 'baseline' blood tests results alluded to in 116 to 124 were taken. In the methods it is stated that blood tests were simply taken "before the transplant". Is that immediately before? 1 day before? 1 week before?

2) It is stated that 2 patients were excluded from LFT analysis post transplant due to "changed liver function as a result of hepatic artery thrombosis". However this is not standard practice to selectively exclude patients just because they had a particular type of complication. Indeed, the authors included many patients who had other complications that could have altered the post-transplant LFTs, and many patients who had IPGF. What is the justification of excluding those with hepatic artery thrombosis? Surely if they had massively abnormal LFTs, it would not matter as the authors are analysing median results, not mean results.

3) It is unclear how many patients ended up having percutaneous liver biopsy. It is stated that 6 had PBC diagnosed via biopsy. How about those who had biopsy but PBC was not diagnosed?

4) The authors found via log-rank testing that APRI >2 (I assume preoperatively) was
associated with poorer overall survival, and that MELD score (I assume preoperatively) was not. However it is not clear from their methods whether they performed log-rank testing for any other variables such as age, gender, Mayo risk score, pre-operative LFTs, post-operative LFTs, Child Pugh score, and if so what the results were.

5) The authors found via log-rank testing that age <50 was associated with greater disease recurrence.However it is not clear from their methods whether they performed log-rank testing for any other variables such as gender, Mayo risk score, APRI score, pre-operative LFTs, post-operative LFTs, Child Pugh score, MELD score and if so what the results were.

6) Similarly, did the authors consider analysing for any more sophisticated transplantation scores such as the UKELD score, MELD-Na score, iMELD score?

Validity of the findings

1) In the results, there are no figures showing the overall Kaplan Meyer survival curves for the whole cohort. This would be meaningful to present in addition the curves stratified by pre-operative APRI score.

2) The authors statement that "protocol biopsies are recommended after transplant" and "In future researches, protocol criteria should be recommended in order to detect early recurrence timely" are contentious. Protocol biopsies of patients after transplant for PBC are rarely routine outside of research setting. I do not know of any major transplant centres who perform this in the absence of LFT dysfunction. How can the authors justify performing so many biopsies on asymptomatic post-transplant PBC patients with acceptable LFTs, given the inherent risks and resource utilization inherent in liver biopsy?

3) The authors state in the discussion that "Consistent with our study’s findings, this previous study showed that a high GGT level at day 7 after liver transplantation is associated with increased survival rates within the first 90 days". This is completely wrong. The authors' study did not analyse the relationship between GGT at day 7 and survival. In fact it did not analyse the relationship between any post-operative LFTs and survival: and in fact it is unclear why the authors have therefore spent so much efforts detailing the trajectory of post-operative LFT changes in their cohort. What is the practical value of detailing these trajectories to the reader of this study? How does it change patient prognosis or management?

---

## Round 0.2 · Major Revisions

Dear Dr. Li,

One reviewer still has concerns regarding your submission. Please make every attempt to address these concerns and resubmit your manuscript.

Reviewer 1 ·

Basic reporting

No additional comments.

Experimental design

No additional comments.

Validity of the findings

No additional comments.

Reviewer 2 ·

Basic reporting

Abstract remains unstructured. By convention, scientific abstracts are separated into sections: aim, methods, results and conclusions.

Abstract states age <50 is a risk factor for PBC recurrence. However now the manuscript states <48 years.


Lines 114-116 do not belong in the methods section. They belong in the results section

"In our study, 26 patients were classified as Child-Pugh grade B; 43 patients, grade C. One patient had a second liver transplant due to thrombosis of the hepatic artery 2 weeks after the first liver transplant."

Experimental design

Nil

Validity of the findings

In the results section, lines 189-192, it is stated that ". Univariate analysis failed to find that age, gender, the MELD score, Child-Pugh score, Mayo risk score , pre- and post- operative liver function were statistically significant in predicting the outcome of liver transplantation." Firstly, this should be "predicting survival", not "predicting the outcome of liver transplantation. Secondly, what type of univariate analysis was this? Was it univariate logistic regression analysis?

Then in the lines 199-203, it is stated that "In a multivariate Cox regression analyse model adjusted for gender, pre-operative Mayo risk score, 201 APRI score, MELD score, a preoperative age of 48 years or younger remained the only independent risk factor of patient recurrence (hazard ratio 0.028, 95% confidence interval 0.01- 0.71, P = 0.03)". Firstly, this should be "disease recurrence", not "patient recurrence". Why was a multivariate Cox regression analysis performed for disease recurrence, and not for survival as above (where only univariate analysis of some kind was performed)?


The authors have stated in a rebuttal that "In our study,7 patients underwent percutaneous liver biopsy postoperatively. Only 1 patient was diagnosed rejection reaction,and another 6 patients were diagnosed recurrent PBC. " This infromation should be included in the study. In fact, in line 189, it states "Of 69 patients who had abnormal liver function, 6 were diagnosed with rPBC via liver puncture biopsy.". This is incorrect: a total of 69 patients were included in the whole study, but it seems only 7 patients had unexplained abnormal liver function that went on to require biopsy.

---

## Round 0.3 · accepted · Accept

Congratulations, I believe that you have addressed the concerns and the manuscript is now acceptible for publication.